# Finding the Hidden Legacies of African American (and Other) Families

**Robert Scott Davis**

Independent Researcher, Blountsville, AL 35031, USA; genws@hiwaay.net

**Abstract:** Documenting and finding lost family history proves as daunting as locating vital statistics, but the effort to perform such a genealogical feat should be undertaken, for good reason, as the resulting information can prove rewarding. This commentary discusses research methods, effective methods, and materials for finding documentation of family history with a special emphasis on African American genealogy. Commonly used strategies are listed with successful examples and recommended sources. The article concludes with an affirmation of the value of research on ancestors beyond just vital records.

**Keywords:** African American; biographical research; family stories

## 1. Introduction

In traditional European American family history, romantic stories abound of ancestors fleeing *Les Miserable*-like circumstances in the Old World, having European nobility in the family tree, coming to America on the *Mayflower*, having a Cherokee princess for an ancestor, or entitlement to a fortune awaiting only legally recognized evidence. Often these tales, however long researched, prove only myths, although the research can turn up unexpected family information. Determining the fact, fiction, and useful leads from legend proves problematic, however.[1]

Alex Haley, in his classic novel *Roots: The Saga of an American Family*, based on his family's oral traditions and his research, represents the common experience in African American family linage for tens of thousands of persons who came to America as enslaved persons and whose descendants moved across what became the United States as the cotton plantation frontier moved south and west. Black family history, however, can prove to be much more for researchers who overcome a dearth of source material of any kind.[2] The problem of lack of records is compounded by female ancestors as in American women were traditionally treated as wards of their fathers or husbands and frequently do not appear at all or not as identified even as named individuals in such sources as the federal censuses before 1850.[3]

The exceptional African American legacies, when found, teach about the dimensions of the whole experience of a people. Even what we are missing for African American families that have survived goes too often overlooked, as illustrated in the epic autobiographies of Black colonial sailor Olaudah Equiano, Revolutionary War era poet Phillis Wheatley Peters, Revolutionary War–War of 1812 soldier James Roberts, and educator Booker T. Washington.[4]

Stories of other times and worlds, and a family's forgotten legacy, can be recovered, although it may seem daunting.[5] On a recent episode of the television show *Finding Your Roots*, white guest celebrity Joe Manganiello learned through a DNA test, for the first time and despite the research that he had done on his family, that he has African American ancestors. Documentation proves more, that he descends from Plato Turner, born in Africa but freed from enslavement by service in the American Revolution.[6] Such searches can also add to the vital records with which research begins, not just "putting meat on the bones





of our ancestors". This journey becomes a great, highly personal adventure that takes the researcher to interesting places and to great libraries.[7]

## 2. Materials

Common goals for a search for the hidden in African American genealogy might include identifying a Native African ancestor; a freed person who escaped from or was released from bondage before general emancipation, 1865–1868; having a patriot who served in the American Revolution; or learning that a relative who was "Buffalo" soldier, that is a member of the segregated Black military units serving from the Civil War to the Korean conflict, in the family. Such research might uncover less well-known but no less important historical events.

Finding such a lost legacy requires knowledge, luck, persistence, and work. The researcher might uncover ancestors who, at least by today's standards, might have led controversial lives.[8] As with all family research, the investigation begins in the family and then uses what tools for research, general and specific, that make more information available.

Watching out for the opportunity to dig deeper must become a habit to be an effective tool.[9] Cultural events and photographs can provide clues to the meaning of information found in family stories.[10] A family story about an alleged relationship with a famous person such as Martin Luther King Jr. or Thomas Jefferson, for example, should start with biographical and genealogical materials on the celebrity, such as those recommended by the staff of the respective historical sites and libraries with the families.[11]

Family historians should always search for the records of collateral ancestral brothers, sisters, and other persons connected to an ancestor, but with an even greater priority when in a situation where records are lacking. Finding living relatives through family trees is posted on such sites as Ancestry.com and Familysearch.org, or even through a search engine such as Google, which can lead to oral histories preserved in some branches of a family but lost in others.

Even if a researcher's ancestor did not serve in the United States Colored Troops in the American Civil War, deposited money in the Freedman's Bank, or gave an interview to the Federal Writer's Project, other relatives, even by marriage, who did might provide valuable information. Indexes to Civil War service and pension records, for example, are easily searched, and the collective information can piece together a family history for generations, even when the researcher must use circumstantial information, such as eliminating other possibilities. Beyond broad online databases such as Ancestry.com, Archivegrid.com, HathiTrust, Internet Archives, JSTOR, Library of Congress Chronicling America, and various other online resources, specific information such as college alumni, craftsmen, physicians, and Roman Catholic ancestors may need to be consulted.[12]

Many hundreds of national sources for biographical sources are available but particularly at major research libraries.[13] The *Biography and Genealogy Master Index* and *Biography in Context*, both by the Gale Corporation, are the most commonly used for biography in general.[14] For African Americans specifically, a general index to biographical information is available.[15] A database reproducing the pages of the sources used is also available at major research institutions.[16] Some major research libraries also have on microfilm the unpublished biographical sketches of Daniel Alexander Payne Murray.[17] Books are available that offer biographical information on specific groups of African Americans, such as Black women writers.[18]

Individual institutions and geographic areas have materials that do not relate to any specific race, such as the index to biographical sketches of prominent Alabama citizens, prepared in 1940, in Southern History of the Birmingham Public Library.[19] William Dollarhide has published a series of books by region and by the state of basic "name lists" to consult.[20] In recent years, however, some libraries have disposed of original card catalogs and other unique paper sources of information that were even listed in published guides, not recognizing that not all of the information they contained is found in online catalogs.

If the research points to a specific geographic area, many libraries, state archives, and other research institutions have guides to their specific resources for African American research, for example, for Edgefield County, South Carolina, or for Western Nebraska.[21] Alabama and Mississippi have published guides, respectively, for Black family research.[22] Such bibliographies and research guides can point to special research tools such as Georgia's index to more than 1400 slave bills of sale as well as post-Civil War apprenticeships and slave births.[23] Similarly, the Historic New Orleans Collection has the Lost Friends database, an online collection of advertisements in the Methodist *Southwestern Christian Advocate* by formerly enslaved persons trying to regain contact with family members sold before emancipation. The ads contain information that extends far across the South beyond Louisiana.[24]

Public libraries can provide historical research assistance concerning their respective areas. A researcher finding Africa as the place of birth in the federal census records of Mobile County, Alabama, for example, might not know that the person listed came to America on the last slave ship, the infamous *Clotilda*, or the connection to the modern community of Africatown that the survivors of that voyage founded. Books have appeared on this ship over the years but only recently have its remains been located. A query to the Mobile Public Library, however, would direct the researcher to relevant sources.[25]

Clues to secret legacies are where and if the researcher can find them. Many of the former Confederate States have 1867–1868 voter lists for male citizens of all races, for example, that, under naturalization, will list "Africa" as the place of birth. Such information can serve as a beginning to more., such as Thomas Honeyblue of Washington County claiming his citizenship in the Alabama state voter registration of 1867 "by one hundred twenty-five years residence in America" since his birth, he records as in Africa.[26] Similarly, in the Chatham County, Georgia 1867 voter returns, Jim Barefield was shown as born in Africa and as "Wanders cargo", meaning to a local expert that he was one of the 400 enslaved people illegally imported on the infamous ship *Wanderer* in one of the last attempts to bring enslaved persons from Africa into the United States in 1858.[27]

A hunt for an African American family's lost past may involve such unusual records, research in more than just the census, court, and other standard records, even in records of other groups such as Native Americans, Jewish families, and Anglo-European lineages.[28] Edward Ball, in reverse of Alex Haley's *Roots*, documented the lives of the enslaved persons owned by his white colonial South Carolina ancestors through his family papers and then set out to successfully find their descendants and to provide them with a family history they surely never imagined.[29]

Some sources omit references to African Americans, however, even when they are present in the records. In some instances, such as the index to Georgia voter registrations of 1857 and the publication of those records for Mobile County, Alabama, the exclusion is explained in the titles of the publications and thus provides direction to African American researchers to check in the original records for the people excluded from the publications.[30]

Omissions are not explained in some other sources, however. The Franklin Garrett Atlanta Necrology of abstracts of records of Atlanta dead, for example, includes only white persons for the 1840s to the 1920s.[31] In Grace G. Davidson, *Early Records of Georgia: Wilkes County*, for another example, slave ownership from a 1773–1774 land court record is omitted without any disclaimer.[32] In that transcription, Austin Dabney, a famous Black Revolutionary War soldier, is surely one of the omitted numbers of enslaved persons in the household of Richard Aycock, which would conflict with the story of Dabney where he was a member of Aycock's household, enslaved under false pretenses by Aycock.[33]

Often research will raise questions that will remain unanswered, sometimes hauntingly so. African-born free person Fenda Freeman, for example, appears in Henry David Thoreau's famous work on liberty *Walden; or, Life in the Woods*.[34] Could she also be the Fenda Lawrence who came to Georgia as a free person in 1772 with her children by her partner in life and in the trade of enslaved labor Irishman Stephen Deane?[35]

### 3. Methods

Discovering the lost past of African American families has traditional methods. Most of the once-enslaved people interviewed by the United States Work Projects Administration's Federal Writers Project in the 1930s, for example, adopted the surnames of their former owners. Many exceptions exist to that practice, even among siblings, but it provides a place to start.[36] A search for information by the possible owners' surname could lead a researcher to colonial Georgia mariner Andrew Elliott. His estate records show that he left property to his granddaughter Isabella, daughter of free African Sylvia Elliott of Gambia.[37] The paper trail could also lead to the epic story of the descendants of George Galphin, a merchant who traded with Native Americans and fathered African American, Native American, and white children.[38]

Sources to be checked using surnames include the work performed by Paul Heinegg to collect thousands of personal histories of free and freed African Americans in Delaware, Maryland, North Carolina, South Carolina, and Virginia before the Civil War. Michael Ports has published the records of free Black registers for Georgia. Registers of persons settling in Georgia with enslaved people document the arrival of white and Black families.[39] Alex K. Patterson has begun a similar series of books of information on enslaved persons and owners, beginning in 1750.[40]

Genealogical records of Black Americans were often not created, but there are also special sources of value in this research. Often these materials serve family history better than as a source for genealogical data. The interviews of the last people enslaved, prepared by the Federal Writers Project in 1936–1938, are among the most famous of these resources and even include valuable personal and personable information on white families.[41] Records of the federal government's Bureau of Refugees, Freedmen, and Abandoned Lands (the "Freedman's Bureau") tell of the lives of individuals and families.[42] The separate Freedman's Savings and Trust Company records are oriented on the individual; however, the information can help in tracing the past places where a family lived, which thus opens other potential sources.[43]

Federal records, in general, although with some exceptions and limitations, document African American history as with all Americans.[44] The United States military, for example, did not fully integrate until 1954, but, with some exceptions, the records were the same but separate from those for white veterans.[45] Works exist for specific soldiers and conflicts.[46] Federal land records after the Civil War document Black history, such as Major Reid as the founder of the Marriott Creek Colony in Cullman County, Alabama, one of the few surviving African American communities formed as African Americans received general emancipation immediately after the Civil War in the period called Reconstruction.[47]

Similarly, federal records relating to Native Americans prove valuable for African American family research, supported by DNA research that has uncovered that African Americans are more likely to have Native American Ancestry than white Americans![48] Claims made of Cherokee and Creek Indian heritage in such federal records as the Guion Miller claims to compensation for Cherokee lands provide a rich source of otherwise non-existent genealogical information on persons, even when of non-Native American descent.[49] Checking these sources for the "white" Tankersley family of Lumpkin County, for example, leads to the colorful local history of this mixed-race Cherokee or mixed-race African American family of Lumpkin County, Georgia. Their racial heritage remains to be resolved by DNA testing.[50]

### 4. Possibilities

Statistically, the standard African American family history should have much in common with the portrayal of a family in Alex Haley's *Roots*, but many exceptions should also exist. How often an African American family has a past waiting that could now be recovered by diligent research cannot be quantified. It would likely be a high percentage, however, as many people today descend from so few individuals in the past. The persons

descending from those people multiplied over time means that additions to the standard family history should be considered for any family.[51] Even in the novel *Gone With the Wind*, the author's famous attention to historical detail provides examples of now little-known African American family history. The enslaved character Prissy, for example, had Cherokee ancestors and earned the privilege of wearing a white headdress, symbolizing that she could take adult responsibility. Scarlett O'Hara had French-Haitian refugees as ancestors, as did the author Margaret Mitchell. They would have come to America bringing with them their enslaved servants.[52]

Historical records survive even when the names of individuals then present do not, at the least, present possibilities but especially with new methods such as DNA testing. The first European settlement in North America, Lucas Vasquez de Ayllon's San Miguel de Gualdape, was made on the South Carolina coast in 1526. It ended when its African slaves revolted and likely joined the Native Americans.[53] Many tri-racial groups in the South, such as the Lumbee and the Melungeon, have stories of ancestors in America before the English Jamestown colony. Georgia, as a colony and state, fought several armed conflicts with neighboring Spanish East Florida over enslaved persons who fled bondage to live at liberty in Spanish territory in Florida and Mexico.[54]

Similarly, likely 7000 African Americans served in the American military during the Revolutionary War, including the first casualty and the first person to die in that conflict, Crispus Attucks, at the Boston Massacre in 1775. George Washington's army would have ceased to exist without his steadfast African American core of soldiers. Many of the rebelling colonies offered emancipation to the enslaved men who served, with their respective masters' permission, until the end of the war. Freedmen and men falsely claiming freedman status served in the American cause.[55] Unfortunately, the names of only a few of these patriots are documented.[56]

African Americans also escaped to freedom by reaching the British lines, where some of them served as soldiers for the King and, from there, left for other lands. Enslaved people of white Loyalist Americans also left, with their owners, to other lands from Europe to the Pacific Islands. The Black loyalist emigres included followers of George Liele, David George, and Henry Washington (the last formerly owned by George Washington). These leaders brought with them the ideals of the American Revolution and contributed to the end of slavery in the British Empire in 1834, an important chapter in how the struggle for emancipation in America also became a history of freedom for all Americans.[57] Some of the British planters returned to America after the war, bringing back their enslaved persons to the new United States![58] Haitian and other soldiers from the Caribbean served with the French army at Savannah and elsewhere in the American Revolution. Some of their numbers deserted and remained in America. The French and Haitian revolutions led to many white French families, with their enslaved people, immigrating to America.[59]

Escaping from bondage had many different sides than just the famous Underground Railroad that helped enslaved persons to escape to the North and, even for free Black persons in the racially restricted northern states, to Canada.[60] Some biographies of African Americans or their respective family histories begin with newspaper notices of individual "runaways" from bondage.[61] Enslaved African Americans lived among the Native Americans inspiring such stories as that of the "Blackfeet" tribe in the South, still an ethnic mystery. Native Americans also owned and sometimes emancipated enslaved Black people.[62]

Some of the men serving in Black Loyalist military units in Georgia and South Carolina did not leave with the British army but remained in America, sometimes forming or joining maroon colonies in the wilderness with escaped slaves, a tradition going back to colonial times. Thousands of African Americans self-emancipated, and some of them did not leave the South but lived with the assumed identity of "free persons of color" or even white.[63]

Enslavement, freedom, and liberty meant different things depending upon the individual, such as for persons not in bondage but also not granted the civil rights of white citizens. In many parts of America, African Americans not in bondage were not allowed to

compete with whites for jobs or land. Thomas Sims of Savannah, for example, belonged to a community of officially enslaved persons but who led independent lives to avoid laws prohibiting free Black labor. He eventually escaped even that level of bondage over problems with his wife to become one of the first victims of the Fugitive Slave Law of 1850, legislation the United States Congress passed that sought to stop enslaved people from escaping to freedom in Canada and Mexico.[64] In pre-Civil War Atlanta, the enslaved workers of trader in enslaved persons Ephraim Ponder operated a factory and worked as independent contract laborers. One of the members of this community, Henry Ossian Flipper, became the first Black graduate of the United States Military Academy. This form of officially enslaved but largely independent Black labor became increasingly common before the Civil War. Cities such as Mobile and New Orleans had Creole classes that had liberties and statuses not available to African Americans in other parts of the country.[65]

During the American Civil War, African Americans served on both sides. Even Confederate General Robert E. Lee's army would not have lasted without African American labor.[66] Black soldiers in the U.S. Army were among the first casualties of the war. The federal government drafted, enlisted, and impressed Black boys and men into the United States Colored Troops.[67] White federal soldiers who escaped Confederate prisons depended upon enslaved people for help in returning to the federal lines. Unfortunately, the names of these enslaved African Americans who helped white soldiers to escape to freedom are seldom, if ever, named in the soldiers' memoirs.[68] Federal armies advancing across the slave states impressed property from Black owners, as well as white, and all could file informative claims for compensation after the Civil War.[69]

African American history is a series of great migrations covering the whole of their collective experience. Aside from the people who left with the British at the end of the American Revolution and the American Colonization Society's work in transporting former slaves to Liberia, some former members of the United States Colored Troops resettled in Kansas and Oklahoma after the Civil War in the West as Exodusters, leaving federal land records. African Americans would also leave homes in the South in the hope of better lives in the Great Migration of modern times, where between 1910 and 1970, 6 million African Americans left poverty and injustice of the southern United States for new opportunities in the states of the North and West, stories for which some people live to tell even now.[70]

## 5. Conclusions

American genealogical research can deteriorate into only filling in a puzzle of relationships with vital records. When family history is sought, sometimes researchers pursue exaggerated or even false stories. Some of our past is always lost, but when it is found, even as much by accident as design, the researcher should make the most of the opportunity to better understand the individuals and the families within their times.

Genealogists must never make the mistake of too many historians, that is, to lose sight of the great potential in family history research for finding special meaning in relationships. For African Americans, that requires more patience, for while the amazing and the exceptional did happen, finding the records can be daunting. History can survive to inspire and to teach, but most of all, to be told and remembered.

**Funding:** This research received no external funding.

**Institutional Review Board Statement:** Not applicable.

**Data Availability Statement:** Not applicable.

**Conflicts of Interest:** The author declares no conflict of interest.

## Notes

1    (Davis 1995). Works available, among others include, (Hite 2013; Keating 2002; Smolenyak 2012; Symes 2017).

2    See (Hernig and Westfield 2014). For historical background on the African American experience, see among other works (Proenza-Coles 2019).

3    Books for specifically researching female relatives include (Carmack 1998; Emm 2019; Schoefer 1999). The encyclopedias of African American women include, among other works, (Hine and Brown 1993; Davis 1982); also see (Berry and Gross 2020).

4    (Equiano 1789; Gates 2003; Roberts 1858; Washington 1901).

5    See (Gates 2009).

6    See (Stern 2023).

7    Some successes in this research that educate and inspire include (Franklin and Schweninger 2006; Torrey and Green 2021; Tomlinson 2014; Weincek 2000; Webster 2023; White 2021).

8    See for example, (Newton 2002).

9    Online, see the Library of Congress "African American Genealogical Research": https://guides.loc.gov/african-american-genealogical-research (accessed on 13 July 2023). Published guides, among other works, include (Jackson 2007; Green 2012; Pinnick 2014; Richard 2019; Streets 2008; Ham 1984).

10    See (Kelbaugh 2022; Gay 2006).

11    For African American connections to Jefferson, a researcher might start with Wiencek (2012).

12    Some examples of databases that are not limited by race include the AMA Deceased Physicians Card file: https://www.nlm.nih.gov/hmd/genealogy/ama-deceased-physicians.html (accessed on 13 July 2023); Museum of Early Southern Decorative Arts (MESDA) Craftsman Database: https://mesda.org/research/craftsman-database/ (accessed on 13 July 2023); and records of execution and lynching in the M. Watt Espy Papers, 1730–2008, M. E. Greenlander Department of Special Collections and Archives, University at Albany, State University of New York. The Library of Congress has begun digitizing and indexing the published credit reports of the Bradstreet and the R. G. Dun companies, 1859–1924: https://www.loc.gov/collections/dun-and-bradstreet-reference-book/about-this-collection/?loclr=blogadm (accessed on 13 July 2023).

13    See "United States Biography": https://www.familysearch.org/en/wiki/United_States_Biography (accessed on 13 July 2023); Wikipedia's incomplete "List of Biographical Dictionaries": https://en.wikipedia.org/wiki/List_of_biographical_dictionaries (accessed on 13 July 2023); and "Library of Congress Bibliographies, Research Guides, and Finding Aids": https://www.loc.gov/rr/program/bib/index.html (accessed on 13 July 2023).

14    The Biography and Genealogy Master Index (BGMI), https://www.gale.com/c/biography-and-genealogy-master-index (accessed on 13 July 2023) is also included in the subscription website Ancestry.com/Ancestrylibrary.com (accessed on 13 July 2023) as is American Genealogical-Biographical Index (AGBI). Another biography bibliography is American Biographical Archives (New York: K. G. Saur, 1986–1990), microfiche.

15    Randall K. Burkett, Nancy Hall Burkett, and Henry Louis Gates Jr., *Black Biography 1790–1950: A Cumulative Index*, 3 vols. (Alexandria, VA: Chadwyck-Healey, 1991). The supplement to Black Biography has an online listing of names and biographies, as well as a bibliography of sources used in these works: https://media2.proquest.com/documents/black_bio_supp.pdf (accessed on 13 July 2023).

16    "African American Biographical Database, 1790–1950": https://proquest.libguides.com/africanamericanbios (accessed on 13 July 2023).

17    Daniel Alexander Payne Murray Papers, 1881–1966, State Historical Society of Wisconsin, Madison: https://digicoll.library.wisc.edu/cgi/f/findaid/findaid-idx?c=wiarchives;view=reslist;subview=standard;didno=uw-whs-micr0577 (accessed on 13 July 2023).

18    See, for oher examples, (Foner 1996; Knight 2019; Ross 2019; Swarns 2023).

19    WPA Index to Alabama Biography: http://bpldb.bplonline.org/db/biographies (accessed on 13 July 2023).

20    See, for example, Dollarhide (2016, 2013).

21    See for example (Williams 2017; Coleman and Hopkins 2022).

22    (Taylor and Rose 2008; Webster 2001).

23    AfriGeneas has this index: https://www.afrigeneas.com/library/ga-slavebills/ (accessed on 13 July 2023).

24    See Lost Friends Advertisements in the Southwestern Christian Advocate: https://www.hnoc.org/database/lost-friends/index.html (accessed on 13 July 2023).

25    Local History & Genealogy, Mobile Public Library, recommends the Clotilda Descendants Association: https://theclotildastory.com/ (accessed on 13 July 2023) and, among other books, (Diouf 2009; Hurston 2018).

26    (Davis 1994). The site Ancestry.com/Ancestrylibrary.com (accessed on 13 July 2023) includes the 1867 voter lists for Alabama, Arkansas, Florida, Georgia, and Texas.

27    For the history of the Wanderer, see (Calonius 2006).

28      Suellen Ocean has published a seven-volume series on discovering her family's multi-ethnic and racial connections, including through DNA research. See (Ocean 2009–2018).

29      See (Ball 1998).

30      See (Brandenburg and Worthy 1995; Rowe 1867).

31      The Franklin Garrett Atlanta Necrology, http://garrett.atlantahistorycenter.com/ (accessed on 13 July 2023), has a separate online index and scanning on Familysearch.org as Cemetery record Atlanta, 1874–1932: https://www.familysearch.org/search/catalog/2139428?availability=Family%20History%20Library (accessed on 13 July 2023).

32      (Davidson 1932). A complete transcript of the now lost Ceded Lands journal is James A. LeConte, "Transcript Records of the Court of Land Commissioners 'Ceded Lands' Later Wilkes, Now Part of Records of Greene Co. Augusta, GA November 19th 1773" (1910), microfilm drawer 154, box 65, and the remains of the original manuscript Ceded Lands journal, 1773–1775, fragments, box OTH-176, Georgia Archives, Morrow.

33      See (Davis 2014).

34      See (Trent 2021).

35      See (Davis 2013).

36      See (Foster 2022; Thomas 1998).

37      See (Pressly 2012).

38      (Hicks and Taukchiray 1998). Further research on Archivegrid.org (accessed on 13 July 2023), would then point the researcher to the Theresa M. Hicks Papers, 1967–2005, in Special Collections, South Caroliniana Library, University of South Carolina, Columbia, South Carolina.

39      (Heinegg 2001–2021; Ports 2015–2016). For persons settling in Georgia and elsewhere in the southeastern United States with ennslaved persons see (Potter 1990; Wilson 2012; AGS Historical Records Committee 2023).

40      See (Patterson 2018).

41      Several publications have explored the Works Ptrojects Administration interviews done in the 1930s of last of the once enslaved Americans. For access to the interviews, now in the collections of the Library of Congress, see Born in Slavery: https://www.loc.gov/collections/slave-narratives-from-the-federal-writers-project-1936-to-1938/about-this-collection/ (accessed on 13 July 2023). The interviews have also been published by state, see Slave Narratives: https://www.loc.gov/item/41021619/ accessed on 13 July 2023. For an example of these narratives used in research see (Hallman 2013).

42      The Freedmen's Bureau records are accessible on Familysearch.com, see "The Freedmen's Bureau": https://www.archives.gov/research/african-americans/freedmens-bureau (accessed on 13 July 2023); also see (Lawson 2019).

43      Ham, Black History: A Guide to Civilian Records in the National Archives, 189–92. The records of the Freedman's Bank are transcribed on Familysearch.org (accessed on 13 July 2023), see "United States, Freedman's Bank Records": https://www.familysearch.org/en/wiki/United_States,_Freedman\T1\textquoterights_Bank_Records_-_FamilySearch_Historical_Records (accessed on 13 July 2023).

44      For research in federal records see Guide to Genealogical Research in the National Archives (Washington: National Archives and Records Administration, 2006).

45      Broad works include (Buckley 2002; Lanning 2002; Weir 2004).

46      See, for example, (Bielakowski 2021; Lanning 2002; Lusted 2014).

47      Allen Reid, General Land Office Records, Bureau of Land Management: https://glorecords.blm.gov/default.aspx (accessed on 13 July 2023); (Sterling 2019); and and for homesteaders see Berenice Alexander Bennett, Black Homesteaders of the South (Charleston, SC: The History Press, 2023)

48      See (Zimmer 2016; Bryc et al. 2015). For African American Research in Native American ancestry see (Walton-Raji 1993).

49      See (Edington and Buswell 1998).

50      See Davis (1999, 1993).

51      See, among other works, (Crandall 1996; Jacobs 2017).

52      (Lippman 1998). Also see (Dickey 2014).

53      See (Hoffman 2004).

54      See (DeMarce 1992; Jennison 2012).

55      See (Davis 2023).

56      See (Crowder 2019; Moss and Scroggins 2004; Heinegg 2021; National Society of the Daughters of the American Revolution 2001). A supplement to the latter work is available at https://www.dar.org/library/research-guides/dar-publications (accessed on 13 July 2023).

57      See (Proenza-Coles 2019; Fischer 2022; Horne 2014).

58      See (Jasanoff 2011; Nash 2006; Gilbert 2012; Kaplan and Kaplan 1989). For free African American evacuees see (Hodges 1996).

59      See (DuBois 2004; Tucker 2023; Langley 1996; Geggus 2002).

60    See (Franklin and Schweninger 1999; Snodgrass 2008). The National Underground Railroad Center in Cincinnati, Ohio is also a valuable resource: https://freedomcenter.org/ (accessed on 13 July 2023).

61    References to an African American Ancestor may appear, for example, in Lathan A. Windley's series of books on pre-1790 notices of escaped slaves in Georgia, Maryland, North Carolina, South Carolina, and Virginia. Online newspaper databases such as the Library of Congress Chronicling America: https://chroniclingamerica.loc.gov/ (accessed on 13 July 2023) identify runaway notices and other records. Chronicling America also includes a bibliography of digitalized newspapers. Also see the Digital Library on American Slavery: http://dlas.uncg.edu/ (accessed on 13 July 2023).

62    (Krauthamer 2013); and, as an example, (Davis 2015).

63    (Delbanco 2018; Franklin and Schweninger 1999); Chris Nordman, "Tracing Free Persons of Color in the Antebellum South: A Selective Bibliography": http://findingafricanamericanancestors.weebly.com/uploads/1/1/8/8/11883350/tracing_fpc_selective_bibliography.pdf (accessed on 13 July 2023).

64    See (Parker 1852).

65    See (Hirsch and Logssdon 1992; Andrews and Higgins 1974).

66    Some important works include (Moss 2008–2019; Rodriguez 2022). For background see (Humphreys 2008; Mendez 2019; Pinheiro 2022).

67    See (Dollarhide 2009; Munden and Beers 1986; Secret 2007; Weidman 1997).

68    See (Davis 2007).

69    (Davis 1991). To search for a Civil War claim filed against the federal government see (Mills 1994; Munden and Beers 1986); and the "Name and Subject Index to Records Used in the Settlement of Claims, 1861–1909", Entry 366, Record Group 56 Records of the Department of the Treasury, National Archives and Records Administration (NARA), National Archives II, College Park, MD.

70    See (Painter 1992; Wright 2022).

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
