# Peer review of "Finding the Hidden Legacies of African American (and Other) Families"

_genealogy, doi:10.3390/genealogy7030046_

Round 1

Reviewer 1 Report

I enjoyed reading this lovely article that describes the importance of explictly looking more broadly when researching family history. Well done. 

Author Response

Thank you!

Reviewer 2 Report

This short article contains a wealth of useful information for those researching African-American families. It may not be particularly new but the material is drawn together in an interesting way, with a focus on exploring resources widely and examining context from a range of different angles. Not all readers however will be versed in US history, so some of the events/organisations etc mentioned - and not explained - make the article sometimes difficult to follow for a non-American.

The footnotes are extensive and some of the information contained in them might be better included in the article, to clarify and explain specific claims. Also the footnotes do not seem to be presented in standard form - but this can presumably be fixed via editing (also some typos and oddly phrased sentences that need editing).

Some specific comments:

Line 136: What is the Federal Writers' Project? (Please explain on first mention)

Line 77: What specific groups? Examples?

Line 113: Why is Edward Balls' action 'in reverse' of Haley?

Line 126: Explain how the stories conflict

Lines 128-132: Not clear what is being said in the 'Fenda' story. Perhaps omit? It is all rather speculative.

Line 166: What is "Reconstruction"

Line 168-169: Odd sentence - rework for clarity

Line 179: Who are Heinigg and Ports?? Also this is a very hard to follow sentence.

Line 197: What was the 'Patriot cause'?

Line 236: What is the 'Fugitive Slave Law of 185'? (something missing from this sentence?)

Lines 251-252: Not sure of the meaning here - rework sentence for clarity

Line 260: What is the 'Great Migration of modern times'?

Line 259: Why an exclamation mark?

I think the article would be clearer and even more informative with some or all of the above issues attended to, and the Conclusion made sharper and clearer.

Some overly long sentences, and some are unclear as mentioned previously. Important to explain some of the terminology, so that the article is accessible to readers not steeped in US history but nevertheless interested in mixed African heritage.

Author Response

Thank you